# KFNN: K-Free Nearest Neighbor For Crowdsourcing

**Wenjun Zhang**
School of Computer Science
China University of Geosciences
Wuhan 430074, China
wjzhang@cug.edu.cn

**Liangxiao Jiang**[*]
School of Computer Science
China University of Geosciences
Wuhan 430074, China
ljiang@cug.edu.cn

**Chaoqun Li**
School of Mathematics and Physics
China University of Geosciences
Wuhan 430074, China
chqli@cug.edu.cn

## Abstract

To reduce annotation costs, it is common in crowdsourcing to collect only a few noisy labels from different crowd workers for each instance. However, the limited noisy labels restrict the performance of label integration algorithms in inferring the unknown true label for the instance. Recent works have shown that leveraging neighbor instances can help alleviate this problem. Yet, these works all assume that each instance has the same neighborhood size, which defies common sense. To address this gap, we propose a novel label integration algorithm called K-free nearest neighbor (KFNN). In KFNN, the neighborhood size of each instance is automatically determined based on its attributes and noisy labels. Specifically, KFNN initially estimates a Mahalanobis distance distribution from the attribute space to model the relationship between each instance and all classes. This distance distribution is then utilized to enhance the multiple noisy label distribution of each instance. Subsequently, a Kalman filter is designed to mitigate the impact of noise incurred by neighbor instances. Finally, KFNN determines the optimal neighborhood size by the max-margin learning. Extensive experimental results demonstrate that KFNN significantly outperforms all the other state-of-the-art algorithms and exhibits greater robustness in various crowdsourcing scenarios. Our codes and datasets are available at https://github.com/jiangliangxiao/KFNN.

## 1 Introduction

Crowdsourcing provides a more cost-effective way to obtain annotated instances than traditional expert annotation [1]. Through crowdsourcing platforms such as Figure Eight and Clickworker, instances can be annotated by crowd workers at a low cost [2, 3]. While more affordable, these workers possess less expertise than domain experts and are more prone to assigning noisy labels to instances [4]. To address this issue, the concept of *repeated annotation* is introduced and becomes popular in crowdsourcing [5]. With *repeated annotation*, each instance is annotated by several workers, thereby obtaining multiple noisy labels. To train supervised models using multiple noisy labels, two main categories of methods have been developed: one-stage methods and two-stage methods. One-stage methods [6, 7, 8] train models directly using multiple noisy labels. Two-stage methods [1, 9] first infer the unknown true label for each instance from its multiple noisy labels via

---

[*]Corresponding author

38th Conference on Neural Information Processing Systems (NeurIPS 2024).

label integration (also known as answer aggregation or ground truth inference) [10] and then train models on integrated labels. One-stage methods, although end-to-end, can only be used to train specifically designed models. As a result, label integration, which is required for the more common two-stage methods, has received a great deal of attention from researchers.

It has been theoretically demonstrated that, when worker annotation is more accurate than random annotation, the more noisy labels an instance receives, the easier it becomes to infer its unknown true label [11]. However, to reduce annotation costs, only a few noisy labels can be collected for each instance in crowdsourcing. The limited noisy labels restrict the performance of label integration algorithms in inferring the unknown true label for the instance. Furthermore, some common strategies in crowdsourcing, such as worker modelling, worker elimination and task assignment [12], fail to mitigate the effects of limited labels in label integration. To alleviate this problem, recent works have begun to focus on leveraging neighbor instances [13, 14, 1]. These works successfully improve the performance of label integration by leveraging the information from neighbor instances obtained by the K-nearest neighbor (KNN) algorithm. However, due to the use of KNN, these algorithms all assume that each instance has the same neighborhood size. This assumption is difficult to hold because it defies common sense, e.g. instances close to the center of classes should have more neighbors than instances close to the boundary of classes.

To address this gap, we propose a novel label integration algorithm called K-free nearest neighbor (KFNN). In KFNN, the optimal neighborhood size of each instance is automatically determined based on its attributes and noisy labels. Notably, KFNN is different from some supervised works [15, 16] that determine the optimal K-value for KNN. Unlike in supervised learning, the true label of each instance in crowdsourcing is unknown and only its multiple noisy labels can be used, which makes it difficult to model the relationship between the instance and all classes. To do this, KFNN initially estimates a Mahalanobis distance distribution from the attribute space to model the relationship between each instance and all classes. This distance distribution is then utilized to enhance the label distribution for each instance. Subsequently, a Kalman filter is designed to mitigate the impact of noise incurred by neighbor instances. Finally, KFNN determines the optimal neighborhood size by the max-margin learning. In general, the contributions of this paper can be summarized as follows:

- We reveal the limitations caused by fixing the neighborhood size in existing label integration algorithms and propose a novel algorithm called KFNN. In KFNN, the neighborhood size of each instance is automatically determined based on its attributes and noisy labels.

- We estimate a Mahalanobis distance distribution from the attribute space to model the relationship between each instance and all classes. This distance distribution enhances the multiple noisy label distribution of each instance.

- We design a Kalman filter to mitigate the impact of noise incurred by neighbor instances and then determine the optimal neighborhood size by the max-margin learning, which provides strong theoretical support for our algorithm.

- Extensive experimental results demonstrate that KFNN significantly outperforms all the other state-of-the-art label integration algorithms and exhibits greater robustness than existing algorithms in various crowdsourcing scenarios.

## 2 Related work

Depending on whether neighbor instances are leveraged or not, existing label integration algorithms can be divided into two categories. The first category of algorithms does not leverage neighbor instances, which considers only the information of the instance itself or the information of all instances globally in label integration. For example, [17] models the ability of each worker with a confusion matrix. In this matrix, each element reflects the probability that this worker annotates an instance with the class corresponding to the row as the class corresponding to the column. [18, 19] are Bayesian versions of [17], which can be used for binary tasks and multi-class tasks, respectively. Further, [20, 21] improve [19] by introducing the correlation between workers. [11, 22, 23] are classical algorithms based on majority voting and they tend to use the label with the highest number of votes as the integrated label. [24, 25, 26] synchronously model the ability of workers and the difficulty of tasks from different perspectives. [27, 28] use clustering algorithms to divide instances into different clusters from different views, and then map these clusters to different classes. Recently, [29] augments the multiple noisy label distributions of instances as new attributes to the original

attribute space and then learns a classifier on the augmented attribute space to predict the integrated labels of instances. [9] constructs graphs for workers and uses a graph neural network to aggregate multi-order information in label integration.

The second category of algorithms performs label integration by leveraging the information from neighbor instances obtained by the KNN algorithm. For example, [13] proposes to use the labels assigned to the neighbor instances of an instance to augment this instance's multiple noisy labels and use the augmented multiple noisy labels to infer the integrated label of this instance. [14] considers both nearest and farthest neighbors in weighted voting to address class-imbalanced tasks. Further, inspired by label distribution learning [30, 31], given an instance, [1] iteratively absorbs the label distributions of its neighbor instances into its label distribution through label distribution propagation.

While simpler and more efficient, the first category of algorithms are limited in effectiveness because each instance can only obtain few noisy labels. Both experimental results and theoretical analysis demonstrate the effectiveness of the second category of algorithms in leveraging the information from neighbor instances. However, these algorithms all assume a fixed neighborhood size for each instance, which is often unrealistic and thus limits their performance. To further ensure that each instance has a free neighborhood size, this paper proposes a novel label integration algorithm called KFNN. KFNN automatically determines the optimal neighborhood size for each instance based on its attributes and noisy labels, which improves the performance and robustness of label integration.

## 3 Algorithm

In this section, we respond to how to automatically determine the optimal neighborhood size for each instance. First, we present some basic notations in crowdsourcing and then define the problem settings. Subsequently, we introduce our KFNN for label integration.

### 3.1 Preliminary

Let $D = \{(\boldsymbol{x}_i, \boldsymbol{L}_i)\}_{i=1}^N$ denote a crowdsourced dataset, where $N$ is the number of instances, and $\boldsymbol{x}_i$ denotes the $i$-th instance in $D$. $\boldsymbol{x}_i$ can be represented as $\{x_{im}\}_{m=1}^M$. Here, $M$ is the dimension of attributes, and $x_{im}$ denotes the attribute value of $\boldsymbol{x}_i$ on the $m$-th attribute $A_m$. $\boldsymbol{L}_i$ denotes multiple noisy labels of $\boldsymbol{x}_i$, which can be expressed as $\{l_{ir}\}_{r=1}^R$. $R$ is the number of workers and $l_{ir}$ denotes the label of $\boldsymbol{x}_i$ annotated by the $r$-th worker $u_r$. $l_{ir}$ takes a value from a fixed set $\{-1, c_1, \ldots, c_q, \ldots, c_Q\}$, where $Q$ is the number of classes, $c_q$ denotes the $q$-th class and $-1$ indicates that $u_r$ has not annotated $\boldsymbol{x}_i$. Label integration aims to infer an integrated label $\hat{y}_i$ for $\boldsymbol{x}_i$ and minimize the error between $\hat{y}_i$ and the unknown true label $y_i$.

Recent works [1, 13] have shown that leveraging neighbor instances $\mathcal{N}_i = \{\boldsymbol{x}_i^k\}_{k=1}^K$ of $\boldsymbol{x}_i$ can mitigate the restriction of limited noisy labels on the performance of label integration. Here, $\boldsymbol{x}_i^k$ denotes the $k$-th nearest neighbor of $\boldsymbol{x}_i$ and $K$ is the neighborhood size. However, in these works, the value of $K$ is fixed for each instance within the same dataset, which does not make sense. On the one hand, instances closer to the center of a class benefit from a larger $K$, as it enables them to collect more labels from similar instances. Conversely, for instances close to the boundary of classes, a larger $K$ plays a negative role in label integration. On the other hand, using a fixed $K$ can bias algorithms towards the majority class in class-imbalanced datasets, as instances from the majority class are more likely to dominate the neighborhood of instances from minority classes. Therefore, we define the Problem 1 to be addressed in this paper as follows:

**Problem 1.** *Given a crowdsourced dataset $D$, how to automatically determine the optimal neighborhood size $K_i^*$ for each instance $\boldsymbol{x}_i$ with $\{x_{im}\}_{m=1}^M$ and $\{l_{ir}\}_{r=1}^R$ but without $y_i$.*

Problem 1 cannot be treated simply as learning an optimal neighborhood size for the KNN algorithm in supervised learning [15, 16]. This is because the true labels of instances in crowdsourcing are unknown. As a result, $K$ can not be evaluated accurately by supervised metrics such as classification accuracy. Moreover, label integration does not divide the crowdsourced dataset into training, validation and test sets, which means that KFNN has to determine $K_i^*$ immediately when inferring $\hat{y}_i$, rather than with a validation phase.

## 3.2 K-free nearest neighbor algorithm

In this subsection, we propose our KFNN to address Problem 1. We argue that $K_i^*$ should be related to the information from both the attribute space and the multiple noisy label space. Based on this, KFNN divides Problem 1 into two parts: 1) How to fuse the information from the attribute space and the multiple noisy label space? 2) How to determine an optimal $K_i^*$ for $\boldsymbol{x}_i$? Correspondingly, KFNN consists of two components, namely label distribution enhancement and K-free optimization, which are used to address the two parts of Problem 1.

### 3.2.1 Label distribution enhancement

For each instance $\boldsymbol{x}_i$, $\{x_{im}\}_{m=1}^M$ reflects all the information of it in the attribute space and $\{l_{ir}\}_{r=1}^R$ reflects all the information of it in the multiple noisy label space. Inspired by label enhancement (LE) [32, 33], we design a label distribution enhancement (LDE) component for KFNN. LDE recovers a potential label distribution using $\{x_{im}\}_{m=1}^M$, and then enhances the multiple noisy label distribution calculated from $\{l_{ir}\}_{r=1}^R$ by this potential label distribution. Specifically, KFNN first uses majority voting to initialize the integrated label $\hat{y}_i$ for $\boldsymbol{x}_i$ as follows:

$$\hat{y}_i = \underset{c \in \{c_1, c_2, \ldots, c_Q\}}{\arg\max} \; p(c_q | \boldsymbol{L}_i), \tag{1}$$

where $p(c_q | \boldsymbol{L}_i)$ can be calculated as follows:

$$p(c_q | \boldsymbol{L}_i) = \frac{\sum_{r=1}^R \delta(l_{ir}, c_q)}{\sum_{q=1}^Q \sum_{r=1}^R \delta(l_{ir}, c_q)}, \tag{2}$$

Here, $p(c_q | \boldsymbol{L}_i)$ reflects the proportion of labels in $\boldsymbol{L}_i$ that take the value $c_q$. The function $\delta(\cdot)$ outputs 1 if its two parameters are identical, and 0 otherwise. Subsequently, according to $\hat{y}_i$, the crowdsourced dataset $D$ can be divided into $Q$ subsets $\{D_q\}_{q=1}^Q$. The subset $D_q$ contains all instances with initial integrated labels of $c_q$, i.e., $D_q = \{\boldsymbol{x}_i | \hat{y}_i = c_q\}_{i=1}^N$. Then, KFNN calculates a Mahalanobis distance distribution $\{d(\boldsymbol{x}_i, D_q)\}_{q=1}^Q$ as follows:

$$d(\boldsymbol{x}_i, D_q) = \sqrt{(\boldsymbol{x}_i - \boldsymbol{\mu}_q)^T \boldsymbol{C}_q^{-1} (\boldsymbol{x}_i - \boldsymbol{\mu}_q)}, \tag{3}$$

where $\boldsymbol{\mu}_q$ denotes the centroid of $D_q$ and $\boldsymbol{C}_q^{-1}$ denotes the inverse matrix of the covariance matrix of $D_q$. $d(\boldsymbol{x}_i, D_q)$ is the Mahalanobis distance from $\boldsymbol{x}_i$ to $D_q$ calculated in the attribute space. A larger $d(\boldsymbol{x}_i, D_q)$ means that $\boldsymbol{x}_i$ is less likely to belong to $c_q$, conversely a smaller $d(\boldsymbol{x}_i, D_q)$ means that $\boldsymbol{x}_i$ tends to belong to $c_q$. Therefore, $\{d(\boldsymbol{x}_i, D_q)\}_{q=1}^Q$ can be used to model the relationship between each instance and all classes. Based on this, $\{d(\boldsymbol{x}_i, D_q)\}_{q=1}^Q$ can be transformed into a potential label distribution $\{p(c_q | \boldsymbol{x}_i, D_q)\}_{q=1}^Q$ as follows:

$$p(c_q | \boldsymbol{x}_i, D_q) = \frac{max(\{d(\boldsymbol{x}_i, D_q)\}_{q=1}^Q) - d(\boldsymbol{x}_i, D_q)}{max(\{d(\boldsymbol{x}_i, D_q)\}_{q=1}^Q) - min(\{d(\boldsymbol{x}_i, D_q)\}_{q=1}^Q)}, \tag{4}$$

where $max(\cdot)$ and $min(\cdot)$ denote the maximum and minimum values of the set, respectively.

In addition to the potential label distribution, a multiple noisy label distribution $\{p(c_q | \boldsymbol{L}_i)\}_{q=1}^Q$ can also be directly transformed from $\boldsymbol{L}_i$. Different from $\{p(c_q | \boldsymbol{x}_i, D_q)\}_{q=1}^Q$, which learns the potential relationship between instances and classes from the attribute space, $\{p(c_q | \boldsymbol{L}_i)\}_{q=1}^Q$ learns the label distribution reflected by noisy labels from the multiple noisy label space. Finally, KFNN fuses them into an enhanced label distribution $\boldsymbol{P}_i = \{p_{iq}\}_{q=1}^Q$ by averaging as follows:

$$p_{iq} = \frac{p(c_q | \boldsymbol{x}_i, D_q) + p(c_q | \boldsymbol{L}_i)}{\sum_{q=1}^Q [p(c_q | \boldsymbol{x}_i, D_q) + p(c_q | \boldsymbol{L}_i)]}. \tag{5}$$

In this way, the enhanced label distribution $\boldsymbol{P}_i$ can fuse the information from the attribute space and the multiple noisy label space. Therefore, the first part of Problem 1 has been addressed.

### 3.2.2 K-free optimization

After obtaining $\boldsymbol{P}_i$ by label distribution enhancement, KFNN proceeds to determine the optimal neighborhood size $K_i^*$ for $\boldsymbol{x}_i$. First, KFNN calculates the distance between each pair of instances $\boldsymbol{x}_1$ and $\boldsymbol{x}_2$ by:

$$d(\boldsymbol{x}_1, \boldsymbol{x}_2) = \sum_{q=1}^{Q} d(\boldsymbol{x}_1, \boldsymbol{x}_2 | D_q), \tag{6}$$

where $d(\boldsymbol{x}_1, \boldsymbol{x}_2 | D_q)$ can be calculated as follows:

$$d(\boldsymbol{x}_1, \boldsymbol{x}_2 | D_q) = \sqrt{(\boldsymbol{x}_1 - \boldsymbol{x}_2)^T \boldsymbol{C}_q^{-1} (\boldsymbol{x}_1 - \boldsymbol{x}_2)}, \tag{7}$$

Compared to the Euclidean distance, Eq. (6) introduces the label information by calculating the distance between $\boldsymbol{x}_1$ and $\boldsymbol{x}_2$ on each subset $D_q$. According to Eq. (6), we can calculate distances between $\boldsymbol{x}_i$ and all instances in $D$. By sorting these distances we can obtain a neighbor sequence $< \boldsymbol{x}_i^1, \ldots, \boldsymbol{x}_i^k, \ldots, \boldsymbol{x}_i^N >$ for $\boldsymbol{x}_i$. Here, $\boldsymbol{x}_i^k$ is the $k$-th neighbor instance of $\boldsymbol{x}_i$ satisfying $d(\boldsymbol{x}_i, \boldsymbol{x}_i^k) \geq d(\boldsymbol{x}_i, \boldsymbol{x}_i^{k-1})$ when $k$ greater than 1. Then, we calculate the weight $w_{ik}$ for $\boldsymbol{x}_i^k$ as follows:

$$w_{ik} = \frac{\sum_{r=1}^{R} \delta(l_{ir}, l_{ikr})}{\sum_{r=1}^{R} [1 - \delta(l_{ir}, -1)] * [1 - \delta(l_{ikr}, -1)]}, \tag{8}$$

where $l_{ikr}$ denotes the label of $\boldsymbol{x}_i^k$ annotated by the $r$-th worker $u_r$. $w_{ik}$ reflects the proportion of workers assigned the same label for $\boldsymbol{x}_i$ and $\boldsymbol{x}_i^k$. Subsequently, $\boldsymbol{x}_i$ is allowed to absorb the enhanced label distributions of neighbor instances in the neighbor sequence one by one. Let $\boldsymbol{P}_i^k = \{p_{iq}^k\}_{q=1}^{Q}$ denote the label distribution of $\boldsymbol{x}_i$ after absorbing $\boldsymbol{x}_i^k$, which can be updated as follows:

$$p_{iq}^k = \frac{p_{iq}^{k-1} + w_{ik} * p_{ikq}}{\sum_{q=1}^{Q} [p_{iq}^{k-1} + w_{ik} * p_{ikq}]}, \quad k \geq 2, \tag{9}$$

where $p_{ikq}$ denotes the probability value corresponding to $c_q$ in the enhanced label distribution of $\boldsymbol{x}_i^k$. Since the first neighbor instance of $\boldsymbol{x}_i$ is itself, $\boldsymbol{P}_i^k = \boldsymbol{P}_i$ when $k$ is equal to 1.

According to $\boldsymbol{P}_i^k$, KFNN calculates a class margin as follows:

$$\widetilde{\mathcal{M}}_k = max(\boldsymbol{P}_i^k) - sec(\boldsymbol{P}_i^k), \tag{10}$$

where $sec(\cdot)$ denotes the second-largest value of the set. Since the true labels are unknown, Eqs. (5) (7) (8) are all designed based on multiple noisy labels, which lead to that $\widetilde{\mathcal{M}}_k$ contains a degree of noise incurred by neighbor instances. Therefore, KFNN designs a Kalman filter to mitigate the impact of noise in $\widetilde{\mathcal{M}}_k$ as follows:

$$\begin{cases} \hat{\mathcal{M}}_k^- = \hat{\mathcal{M}}_{k-1} \\ \mathcal{P}_k^- = \mathcal{P}_{k-1} + \alpha \\ \mathcal{K}_k = \dfrac{\mathcal{P}_k^-}{\mathcal{P}_k^- + \beta} \\ \hat{\mathcal{M}}_k = \hat{\mathcal{M}}_k^- + \mathcal{K}_k * (\widetilde{\mathcal{M}}_k - \hat{\mathcal{M}}_k^-) \\ \mathcal{P}_k = (1 - \mathcal{K}_k) * \mathcal{P}_k^- \end{cases}, \tag{11}$$

where $\hat{\mathcal{M}}_k$ denotes the filtered margin, determined by both the estimated margin $\hat{\mathcal{M}}_k^-$ and the calculated margin $\widetilde{\mathcal{M}}_k$. The designed Kalman filter can be divided into an estimation phase and an update phase. In the estimation phase, the filter estimates $\hat{\mathcal{M}}_k^-$ and the estimated error $\mathcal{P}_k^-$ based on the filtered margin $\hat{\mathcal{M}}_{k-1}$ and error $\mathcal{P}_{k-1}$ of the previous time index. In the update phase, the filter first updates the Kalman gain $\mathcal{K}_k$ of the $k$-th time index and then updates $\hat{\mathcal{M}}_k$ and error $\mathcal{P}_k$ of the $k$-th time index according to $\mathcal{K}_k$. $\alpha$ and $\beta$ are the process error and the measurement error in the Kalman filter. When $k$ is equal to 0, $\hat{\mathcal{M}}_k$ takes the value of 0 and $\mathcal{P}_k$ takes the value of 1.

To address the second part of Problem 1, KFNN determines the optimal neighborhood size $K_i^*$ for $\boldsymbol{x}_i$ by the max-margin learning as follows:

$$K_i^* = \arg\max_{k \in \{1,2,\cdots,N\}} \hat{\mathcal{M}}_k. \tag{12}$$

Ultimately, according to $K_i^*$, KFNN updates the integrated label $\hat{y}_i$ for $\boldsymbol{x}_i$ as follows:

$$\hat{y}_i = \arg\max_{c \in \{c_1,c_2,\cdots,c_Q\}} \boldsymbol{P}_i^{K_i^*}. \tag{13}$$

The whole learning process of KFNN is shown in Algorithm 1. In Algorithm 1, lines 1-3 initialize the integrated label and multiple noisy label distribution for each instance and their time complexity is $O(NQR)$. Line 4 divides the crowdsourced dataset $D$ into $Q$ subsets and its time complexity is $O(NQ)$. Lines 5-9 perform label distribution enhancement and their time complexity is $O(NM^2Q)$. Line 11 calculates the distances from $\boldsymbol{x}_i$ to other instances and sorts these distances, its time complexity is $O(NM^2Q + N\log(N))$. Lines 12-16 calculate the margins $\{\widetilde{\mathcal{M}}_k\}_{k=1}^N$ and their time complexity is $O(NR + NQ)$. Line 17 filters the margins $\{\widetilde{\mathcal{M}}_k\}_{k=1}^N$ and its time complexity is $O(N)$. Line 18 determines the optimal neighborhood size and its time complexity is $O(N)$. Line 19 infers the integrated label for each instance and its time complexity is $O(Q)$. Therefore, the time complexity of lines 10-20 is $O(N^2(M^2Q + \log(N) + R))$. If only the highest order terms are taken, the time complexity of KFNN is $O(N(NM^2Q + N\log(N) + NR + QR))$.

---

**Algorithm 1** The learning process of KFNN

---

**Require:** $D = \{(\boldsymbol{x}_i, \boldsymbol{L}_i)\}_{i=1}^N$ - a crowdsourced dataset; $\alpha$, $\beta$ - the predefined parameters
**Ensure:** $\{\hat{y}_i\}_{i=1}^N$ - the integrated labels
 1: **for** $i = 1$ to $N$ **do**
 2:     Initialize $\hat{y}_i$ and $\{p(c_q|\boldsymbol{L}_i)\}_{q=1}^Q$ for $\boldsymbol{x}_i$ by Eqs. (1) (2)
 3: **end for**
 4: Divide $D$ into $\{D_q\}_{q=1}^Q$ based on $\hat{y}_i$
 5: **for** $i = 1$ to $N$ **do**
 6:     Calculate $\{d(\boldsymbol{x}_i, D_q)\}_{q=1}^Q$ for $\boldsymbol{x}_i$ by Eq. (3)
 7:     Transform $\{d(\boldsymbol{x}_i, D_q)\}_{q=1}^Q$ into $\{p(c_q|\boldsymbol{x}_i, D_q)\}_{q=1}^Q$ by Eq. (4)
 8:     Fuse $\{p(c_q|\boldsymbol{x}_i, D_q)\}_{q=1}^Q$ and $\{p(c_q|\boldsymbol{L}_i)\}_{q=1}^Q$ into $\boldsymbol{P}_i = \{p_{iq}\}_{q=1}^Q$ by Eq. (5)
 9: **end for**
10: **for** $i = 1$ to $N$ **do**
11:     Calculate $< \boldsymbol{x}_i^1, \ldots, \boldsymbol{x}_i^k, \ldots, \boldsymbol{x}_i^N >$ for $\boldsymbol{x}_i$ by Eqs. (6) (7)
12:     **for** $k = 1$ to $N$ **do**
13:         Calculate the weight $w_{ik}$ for $\boldsymbol{x}_i^k$ by Eq. (8)
14:         Update the label distribution $\boldsymbol{P}_i^k$ by Eq. (9)
15:         Calculate the $\widetilde{\mathcal{M}}_k$ by Eq. (10)
16:     **end for**
17:     Filter $\{\widetilde{\mathcal{M}}_k\}_{k=1}^N$ using the designed Kalman filter by Eq. (11)
18:     Determine the optimal neighborhood size $K_i^*$ for $\boldsymbol{x}_i$ by Eq. (12)
19:     Infer the integrated label $\hat{y}_i$ for $\boldsymbol{x}_i$ by Eq. (13)
20: **end for**
21: **return** $\{\hat{y}_i\}_{i=1}^N$

---

## 4  Theoretical analysis

In this section, we provide some detailed theoretical analysis for KFNN. First, in Eq. (6), KFNN defines the distance $d(\boldsymbol{x}_1, \boldsymbol{x}_2)$ between $\boldsymbol{x}_1$ and $\boldsymbol{x}_2$ based on the Mahalanobis distance $d(\boldsymbol{x}_1, \boldsymbol{x}_2|D_q)$ rather than the traditional Euclidean distance $d_E(\boldsymbol{x}_1, \boldsymbol{x}_2)$. According to Eqs. (3) (7), the Mahalanobis distance works based on a basic assumption, which can be described as follows:

**Assumption 1.** *Given the subset $D_q$, its covariance matrix $\boldsymbol{\mathcal{C}}_q$ is a nonsingular matrix.*

The Assumption 1 holds based on the condition that $|\boldsymbol{\mathcal{C}}_q|$ is non-zero, which is usually satisfied. Even if this condition is not satisfied, we can ensure that the Assumption 1 holds by adding a small value to each element of the principal diagonal on $\boldsymbol{\mathcal{C}}_q$ until $|\boldsymbol{\mathcal{C}}_q|$ is non-zero.

**Theorem 1.** *If Assumption 1 holds, there will be an orthogonal matrix $\mathcal{P}$ satisfying that $\mathcal{P}^{-1}\mathcal{C}_q\mathcal{P} = \mathcal{P}^T\mathcal{C}_q\mathcal{P} = \Lambda$, where $\Lambda$ is a diagonal matrix with all $M$ eigenvalues of $\mathcal{C}_q$ as its elements of the principal diagonal.*

Due to the limited pages, the proof of Theorem 1 is provided in Appendix A. Based on Theorem 1, we can obtain some interesting corollaries about Eqs. (3) (6) (7).

**Corollary 1.** *Compared to $d_E(\boldsymbol{x}_1, \boldsymbol{x}_2)$, $d(\boldsymbol{x}_1, \boldsymbol{x}_2)$ in Eq. (6) does not suffer from the correlation and magnitude of attributes.*

*Proof.* According to Theorem 1, $d(\boldsymbol{x}_1, \boldsymbol{x}_2|D_q)$ can be transformed as follows:

$$
\begin{aligned}
d(\boldsymbol{x}_1, \boldsymbol{x}_2|D_q) &= \sqrt{(\boldsymbol{x}_1 - \boldsymbol{x}_2)^T\mathcal{C}_q^{-1}(\boldsymbol{x}_1 - \boldsymbol{x}_2)} \\
&= \sqrt{(\boldsymbol{x}_1 - \boldsymbol{x}_2)^T((\mathcal{P}^T)^{-1}\Lambda\mathcal{P}^{-1})^{-1}(\boldsymbol{x}_1 - \boldsymbol{x}_2)} \cdot \\
&= \sqrt{(\mathcal{P}^T(\boldsymbol{x}_1 - \boldsymbol{x}_2))^T\Lambda^{-1}(\mathcal{P}^T(\boldsymbol{x}_1 - \boldsymbol{x}_2))}
\end{aligned}
\tag{14}
$$

When $\Lambda^{-1}$ is not considered, the derivation of Eq. (14) implies that $d(\boldsymbol{x}_1, \boldsymbol{x}_2|D_q)$ is the Euclidean distance of instances after orthogonal transformation using $\mathcal{P}^T$. After orthogonal transformation, the attributes are independent of each other, so $d(\boldsymbol{x}_1, \boldsymbol{x}_2)$ does not suffer from the correlation of attributes. $\Lambda^{-1}$ is equivalent to $\text{diag}(\frac{1}{\lambda_1}, \frac{1}{\lambda_2}, \ldots, \frac{1}{\lambda_M})$, where $\lambda_M$ is the $M$-th eigenvalue of $\mathcal{C}_q$ and is equal to the variance on the direction of the corresponding eigenvectors. Briefly, $\Lambda^{-1}$ ensures that the calculated result on each dimension is normalized by the corresponding variance when calculating the distance by Eq. (7). Therefore, $d(\boldsymbol{x}_1, \boldsymbol{x}_2)$ does not also suffer from the magnitude of attributes. $\square$

**Corollary 2.** *Compared to $d_E(\boldsymbol{x}_1, \boldsymbol{x}_2)$, $d(\boldsymbol{x}_1, \boldsymbol{x}_2)$ in Eq. (6) provides a smaller distance for $\boldsymbol{x}_1$ and $\boldsymbol{x}_2$ coming from the same class.*

*Proof.* $\mathcal{P}^T$ causes the original attribute space to be rotated according to the direction of the eigenvectors of $\mathcal{C}_q$, and $\Lambda^{-1}$ causes the rotated attribute space to be scaled according to the eigenvalues $\mathcal{C}_q$. Referring to the principle of principal component analysis [34], the eigenvectors of $\mathcal{C}_q$ reflect the principal component directions of $D_q$. This means that $d(\boldsymbol{x}_1, \boldsymbol{x}_2)$ will provide a smaller distance for instances coming from the same class compared to $d_E(\boldsymbol{x}_1, \boldsymbol{x}_2)$. $\square$

**Assumption 2.** *When we estimate $\hat{\mathcal{M}}_k^-$ based on $\hat{\mathcal{M}}_{k-1}$, the estimated error satisfies N(0, $\mathcal{P}_k^-$). When we measure $\widetilde{\mathcal{M}}_k$ by Eq. (10), the measurement error satisfies N(0, $\beta$).*

The Kalman filter we designed as Eq. (11) works based on Assumption 2, which usually holds because the noise in practice usually satisfies a normal distribution. Since $\hat{\mathcal{M}}_{k-1}$ changes in each time index, the variance of the estimated error $\mathcal{P}_k^-$ changes with the time index. Since Eq. (10) remains constant, the variance of the measurement error $\beta$ is constant. According to Assumption 2, the following theorem can be proved:

**Theorem 2.** *When the Kalman gain $\mathcal{K}_k$ takes the value $\frac{\mathcal{P}_k^-}{\mathcal{P}_k^- + \beta}$, the error between the filtered margin $\hat{\mathcal{M}}_k$ and the true margin $\mathcal{M}_k$ is minimized.*

*Proof.* When Assumption 2 holds, due to $\hat{\mathcal{M}}_k = \hat{\mathcal{M}}_k^- + \mathcal{K}_k * (\widetilde{\mathcal{M}}_k - \hat{\mathcal{M}}_k^-)$, it can be proved that minimizing the error between $\hat{\mathcal{M}}_k$ and $\mathcal{M}_k$ is equivalent to minimizing the variance of $\hat{\mathcal{M}}_k$. Since $\hat{\mathcal{M}}_k^-$ and $\widetilde{\mathcal{M}}_k$ are independent of each other, the following equation can be derived:

$$
\begin{aligned}
Var(\hat{\mathcal{M}}_k) &= Var(\hat{\mathcal{M}}_k^- + \mathcal{K}_k * (\widetilde{\mathcal{M}}_k - \hat{\mathcal{M}}_k^-)) \\
&= (1 - \mathcal{K}_k)^2 * Var(\hat{\mathcal{M}}_k^-) + \mathcal{K}_k^2 * Var(\widetilde{\mathcal{M}}_k)
\end{aligned}
\tag{15}
$$

where $Var(\cdot)$ denotes the variance of the variable. According to Assumption 2, $Var(\hat{\mathcal{M}}_k^-)$ equals to $\mathcal{P}_k^-$ and $Var(\widetilde{\mathcal{M}}_k)$ equals to $\beta$. To minimize the error between the filtered margin $\hat{\mathcal{M}}_k$ and the true

margin $\mathcal{M}_k$, we can calculate the partial derivative $\frac{\partial Var(\hat{\mathcal{M}}_k)}{\partial \mathcal{K}_k}$ as follows:

$$\frac{\partial Var(\hat{\mathcal{M}}_k)}{\partial \mathcal{K}_k} = -2 * (1 - \mathcal{K}_k) * \mathcal{P}_k^- + 2 * \mathcal{K}_k * \beta. \qquad (16)$$

Ultimately, it can be proved that $\mathcal{K}_k$ is equal to $\frac{\mathcal{P}_k^-}{\mathcal{P}_k^- + \beta}$ by setting $\frac{\partial Var(\hat{\mathcal{M}}_k)}{\partial \mathcal{K}_k}$ to 0. $\qquad\square$

**Theorem 3.** *The larger $\hat{\mathcal{M}}_k$ is, the better the corresponding neighborhood size $k$ is.*

Theorem 3 ensures the effectiveness of KFNN in determining the optimal neighborhood size by the max-margin learning, and its proof is provided in Appendix B due to the limited pages.

## 5 Experiments

### 5.1 Experimental setup

To evaluate the effectiveness of KFNN, we construct extensive experiments on the whole 34 simulated and two real-world crowdsourced datasets published on the Crowd Environment and its Knowledge Analysis (CEKA) [35] platform. For simulated datasets, we first use the unsupervised attribute filter *ReplaceMissingValues* in the Waikato Environment and Knowledge Analysis (WEKA) [36] platform to replace all missing values. Subsequently, with the CEKA platform, we hide true labels of simulated datasets and simulate five workers whose label qualities are randomly generated from a normal distribution with N(0.65, 0.05$^2$) to annotate these datasets. The real-world datasets, *Income* and *Leaves*, which were both collected from the online platform Amazon Mechanical Turk (AMT), can be used directly without any processing since they do not contain missing values.

We compare our KFNN with six state-of-the-art label integration algorithms. Among them, MV (majority voting) [11] is the simplest label integration algorithm and is used as a baseline for all algorithms. IWMV (iterative weighted majority voting) [22], AALI (attribute augmentation-based label integration) [29], and LAGNN (label aggregation with graph neural networks) [9] are three state-of-the-art label integration algorithms that do not leverage neighbor instances. LAWMV (label augmented and weighted majority voting) [13] and MNLDP (multiple noisy label distribution propagation) [1] are two state-of-the-art label integration algorithms that leverage neighbor instances. For MV, we use the existing implementation of the CEKA platform. For IWMV, AALI, LAGNN, LAWMV, and MNLDP, we use the implementations provided by their authors. All parameters of the comparison algorithms are set to the recommended values in the corresponding published papers. In addition, since true labels are unknown in our experiments, we use the lazy version of LAGNN. In our KFNN, $\alpha$ and $\beta$ are set to 0.1 and 1 by default.

The performance of each algorithm is evaluated using the Macro-F1 score, which highlights the performance of algorithms on different classes and better reveals algorithmic limitations compared to traditional integration accuracy. Due to the limited pages, more detailed descriptions of the experimental datasets and metrics are provided in Appendix C. All experiments are independently repeated ten times on a Windows 10 machine with an AMD Athlon(tm) X4 860K Quad Core Processor @ 3.70 GHz and 16 GB of RAM, and we report the average results of ten experiments.

### 5.2 Results and discussions

**Simulation experiment results.** Table 1 shows the detailed Macro-F1 score (%) comparisons of each label integration algorithm on each simulated dataset, respectively. Based on these results, we perform the Wilcoxon signed-rank test [37] to further compare each pair of algorithms. Table 3 summarizes the Wilcoxon test results. In Table 3, the symbol ● indicates that the algorithm in the row significantly outperforms the algorithm in the corresponding column, the symbol ○ indicates the exact opposite of that indicated by the symbol ●, and the missing item indicates no significant difference between the algorithm in the row and the algorithm in the column. The significance levels of the lower and upper diagonals are $\alpha = 0.05$ and $\alpha = 0.1$, respectively. Based on these experimental results, we can summarize the following highlights: 1) The average Macro-F1 score of KFNN on all datasets is 79.64%, which is much higher than those of MV (72.46%), IWMV (72.71%), AALI (72.95%), LAGNN (73.71%), LAWMV (73.44%) and MNLDP (76.68%). KFNN achieves the highest Macro-F1

Table 1: The Macro-F1 score (%) comparisons for KFNN versus its comparison algorithms on 34 simulated datasets.

| Dataset | MV | IWMV | AALI | LAGNN | LAWMV | MNLDP | KFNN |
|---|---|---|---|---|---|---|---|
| anneal | 77.61 | 78.42 | 79.30 | 78.41 | 49.89 | 83.69 | 75.00 |
| audiology | 56.44 | 56.52 | 34.30 | 72.19 | 52.73 | 45.17 | 57.29 |
| autos | 81.27 | 81.45 | 73.80 | 81.25 | 77.02 | 77.46 | 78.86 |
| balance-scale | 80.55 | 81.94 | 81.36 | 81.87 | 62.13 | 78.41 | 88.68 |
| biodeg | 66.20 | 66.20 | 70.33 | 69.25 | 77.00 | 73.80 | 75.81 |
| breast-cancer | 65.73 | 65.73 | 64.92 | 66.79 | 55.29 | 59.63 | 57.24 |
| breast-w | 68.92 | 68.92 | 79.92 | 69.91 | 93.58 | 87.45 | 84.90 |
| car | 80.22 | 81.42 | 81.83 | 82.56 | 51.85 | 70.84 | 94.25 |
| credit-a | 72.88 | 72.88 | 76.19 | 74.32 | 87.36 | 78.38 | 81.72 |
| credit-g | 65.42 | 65.42 | 65.23 | 66.81 | 54.92 | 58.10 | 61.91 |
| diabetes | 69.11 | 69.11 | 69.54 | 68.11 | 64.42 | 68.56 | 66.08 |
| heart-c | 74.39 | 74.39 | 74.45 | 74.70 | 85.80 | 79.33 | 75.55 |
| heart-h | 78.50 | 78.50 | 78.58 | 72.07 | 85.58 | 82.98 | 78.80 |
| heart-statlog | 72.81 | 72.81 | 75.43 | 74.42 | 81.88 | 78.25 | 78.10 |
| hepatitis | 54.34 | 54.38 | 57.08 | 55.74 | 60.08 | 66.01 | 61.77 |
| horse-colic | 67.89 | 67.89 | 70.35 | 72.18 | 78.24 | 71.52 | 73.34 |
| hypothyroid | 58.04 | 58.59 | 42.90 | 60.70 | 24.00 | 62.16 | 63.56 |
| ionospheref | 70.61 | 70.71 | 76.38 | 67.16 | 63.69 | 76.91 | 82.31 |
| iris | 81.12 | 81.84 | 87.82 | 81.55 | 98.27 | 97.14 | 97.13 |
| kr-vs-kp | 75.49 | 75.49 | 77.21 | 76.29 | 84.42 | 86.93 | 93.72 |
| labor | 67.93 | 66.42 | 75.48 | 68.01 | 80.84 | 72.16 | 76.84 |
| letter | 93.77 | 94.19 | 95.84 | 94.75 | 98.57 | 99.61 | 99.49 |
| lymph | 69.49 | 68.69 | 56.85 | 71.52 | 64.56 | 59.88 | 70.62 |
| mushroom | 76.06 | 76.06 | 81.97 | 76.19 | 92.69 | 95.63 | 97.98 |
| segment | 89.35 | 90.70 | 92.10 | 90.77 | 96.53 | 98.16 | 98.80 |
| sick | 29.60 | 29.60 | 31.23 | 28.77 | 4.85 | 46.92 | 45.46 |
| sonar | 74.75 | 74.75 | 77.94 | 74.15 | 77.49 | 82.04 | 80.74 |
| spambase | 73.11 | 73.11 | 76.80 | 72.95 | 78.92 | 81.80 | 87.48 |
| tic-tac-toe | 66.84 | 66.84 | 67.37 | 71.75 | 62.60 | 40.70 | 64.50 |
| vehicle | 86.38 | 87.23 | 87.29 | 88.02 | 88.85 | 90.00 | 96.43 |
| vote | 68.69 | 68.69 | 67.18 | 71.02 | 90.83 | 83.07 | 91.29 |
| vowel | 92.48 | 93.07 | 94.23 | 92.35 | 95.23 | 99.81 | 99.05 |
| waveform | 82.13 | 83.70 | 83.04 | 84.52 | 94.88 | 91.88 | 92.49 |
| zoo | 75.50 | 76.34 | 76.02 | 75.08 | 81.93 | 82.64 | 80.60 |
| **Average** | 72.46 | 72.71 | 72.95 | 73.71 | 73.44 | 76.68 | 79.64 |

Table 2: The integration accuracy (%) comparisons for KFNN versus its comparison algorithms on 34 simulated datasets.

| Dataset | MV | IWMV | AALI | LAGNN | LAWMV | MNLDP | KFNN |
|---|---|---|---|---|---|---|---|
| anneal | 84.96 | 85.88 | 86.22 | 85.50 | 85.11 | 91.38 | 89.79 |
| audiology | 78.36 | 78.45 | 78.54 | 78.45 | 79.20 | 78.10 | 81.77 |
| autos | 85.07 | 85.46 | 85.02 | 85.85 | 88.63 | 85.07 | 85.85 |
| balance-scale | 79.25 | 80.32 | 80.35 | 80.29 | 88.82 | 87.86 | 93.49 |
| biodeg | 74.32 | 74.32 | 79.06 | 76.82 | 84.59 | 80.99 | 83.91 |
| breast-cancer | 76.08 | 76.08 | 75.94 | 77.24 | 80.10 | 77.17 | 69.23 |
| breast-w | 76.15 | 76.15 | 87.00 | 77.14 | 95.71 | 90.84 | 87.55 |
| car | 81.46 | 82.31 | 82.92 | 83.28 | 83.23 | 85.90 | 94.34 |
| credit-a | 75.17 | 75.17 | 77.13 | 76.42 | 88.91 | 81.20 | 83.33 |
| credit-g | 75.81 | 75.81 | 76.00 | 77.05 | 79.78 | 75.70 | 69.62 |
| diabetes | 76.37 | 76.37 | 76.64 | 75.40 | 80.43 | 78.55 | 67.33 |
| heart-c | 74.49 | 74.49 | 74.55 | 74.82 | 85.94 | 79.50 | 75.71 |
| heart-h | 79.56 | 79.56 | 79.63 | 73.23 | 86.97 | 84.25 | 80.17 |
| heart-statlog | 75.00 | 75.00 | 78.63 | 76.44 | 84.41 | 81.26 | 79.30 |
| hepatitis | 74.65 | 74.58 | 74.26 | 75.68 | 87.10 | 86.19 | 79.23 |
| horse-colic | 73.94 | 73.94 | 74.51 | 77.74 | 84.51 | 77.47 | 79.92 |
| hypothyroid | 80.32 | 81.53 | 79.28 | 83.24 | 92.29 | 93.27 | 92.40 |
| ionospheref | 77.09 | 77.15 | 79.86 | 73.96 | 80.74 | 85.01 | 85.53 |
| iris | 81.13 | 81.87 | 87.80 | 81.53 | 98.27 | 97.13 | 97.13 |
| kr-vs-kp | 76.30 | 76.30 | 78.47 | 77.13 | 85.70 | 88.18 | 93.87 |
| labor | 75.61 | 74.39 | 78.60 | 75.79 | 87.54 | 82.63 | 83.16 |
| letter | 93.76 | 94.19 | 95.84 | 94.75 | 98.57 | 99.62 | 99.49 |
| lymph | 77.97 | 77.70 | 78.24 | 78.31 | 84.66 | 82.97 | 80.41 |
| mushroom | 76.71 | 76.71 | 84.21 | 76.85 | 93.29 | 95.78 | 98.09 |
| segment | 89.35 | 90.70 | 92.10 | 90.77 | 96.55 | 98.16 | 98.80 |
| sick | 76.87 | 76.87 | 78.34 | 76.28 | 93.98 | 89.81 | 88.78 |
| sonar | 75.91 | 75.91 | 77.02 | 75.19 | 80.00 | 83.41 | 81.78 |
| spambase | 77.49 | 77.49 | 82.01 | 77.33 | 85.50 | 85.78 | 90.15 |
| tic-tac-toe | 74.43 | 74.43 | 75.10 | 78.51 | 78.54 | 71.99 | 74.09 |
| vehicle | 86.39 | 87.23 | 87.29 | 88.03 | 88.97 | 90.09 | 96.43 |
| vote | 74.11 | 74.11 | 77.98 | 75.77 | 92.64 | 86.34 | 93.20 |
| vowel | 92.38 | 92.99 | 94.21 | 92.25 | 95.23 | 99.81 | 99.05 |
| waveform | 82.13 | 83.70 | 82.96 | 84.52 | 94.89 | 91.89 | 92.49 |
| zoo | 80.40 | 81.29 | 87.13 | 81.68 | 91.78 | 91.78 | 86.63 |
| **Average** | 79.09 | 79.37 | 81.26 | 79.80 | 87.72 | 86.33 | 86.24 |

Table 3: The Macro-F1 score (%) comparisons using Wilcoxon tests for KFNN versus its comparison algorithms.

| | MV | IWMV | AALI | LAGNN | LAWMV | MNLDP | KFNN |
|---|---|---|---|---|---|---|---|
| MV | - | ∘ | ∘ | ∘ | | ∘ | ∘ |
| IWMV | ● | - | ∘ | ∘ | | ∘ | ∘ |
| AALI | ● | ● | - | | | ∘ | ∘ |
| LAGNN | ● | ● | | - | | ∘ | ∘ |
| LAWMV | | | | | - | | ∘ |
| MNLDP | ● | ● | ● | ● | | - | ∘ |
| KFNN | ● | ● | ● | ● | ● | ● | - |

Table 4: The integration accuracy (%) comparisons using Wilcoxon tests for KFNN versus its comparison algorithms.

| | MV | IWMV | AALI | LAGNN | LAWMV | MNLDP | KFNN |
|---|---|---|---|---|---|---|---|
| MV | - | ∘ | ∘ | ∘ | ∘ | ∘ | ∘ |
| IWMV | ● | - | ∘ | ∘ | ∘ | ∘ | ∘ |
| AALI | ● | ● | - | ● | ∘ | ∘ | ∘ |
| LAGNN | ● | ● | ∘ | - | ∘ | ∘ | ∘ |
| LAWMV | ● | ● | ● | ● | - | ● | |
| MNLDP | ● | ● | ● | ● | ∘ | - | |
| KFNN | ● | ● | ● | ● | | | - |

score, which indicates that KFNN is more effective and robust than these comparison algorithms in various crowdsourcing scenarios. 2) Among all comparison algorithms of KFNN, MNLDP performs better than IWMV, AALI and LAGNN, which demonstrates the advantages of leveraging neighbor instances. 3) Based on the Wilcoxon test results, KFNN significantly outperforms all comparison algorithms, which strongly validates the effectiveness and robustness of KFNN. Besides, we also observe the experimental results in terms of the integration accuracy, which are shown in Tables 2 and 4. According to Tables 2 and 4, we can see that KFNN can also achieve better or comparable integration accuracy compared with these state-of-the-art label integration algorithms. These results again validate the effectiveness and robustness of KFNN.

**Real-world experiment results.**  Compared to simulated crowdsourced datasets, real-world crowdsourced datasets may include some special factors that influence label integration to work effectively, such as sparsity and bias. Therefore, we further observe the performance of KFNN and its comparison algorithms on two real-world datasets, *Income* and *Leaves*. Figure 1 shows the detailed Macro-F1 score (%) and integration accuracy (%) comparisons of each label integration algorithm on *Income* and *Leaves*, respectively. As can be seen from Figure 1, compared to these state-of-the-art label integration algorithms, our KFNN achieves the highest integration accuracies and Macro-F1 scores on both *Income* and *Leaves*. These results strongly support the effectiveness of our KFNN.

**Parameter sensitivity analysis.**  There are two parameters $\alpha$ and $\beta$ that can be adjusted in the Kalman filter designed by KFNN. To observe the effect of these two parameters on the performance of KFNN, we perform the parameter sensitivity analysis for KFNN on *Income* and *Leaves*. We change both $\alpha$ and $\beta$ from 0.1 to 1 and then observe the Macro-F1 score of KFNN on two datasets.

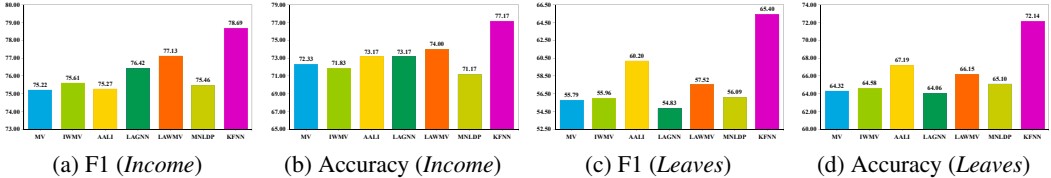

| (a) F1 (*Income*) | (b) Accuracy (*Income*) | (c) F1 (*Leaves*) | (d) Accuracy (*Leaves*) |

Figure 1: The Macro-F1 scores (%) and integration accuracies (%) of KFNN and its comparison algorithms on the *Income* and *Leaves* datasets.

Figure 2a and Figure 2b show the Macro-F1 score of KFNN on *Income* and *Leaves* when $\alpha$ and $\beta$ vary. Based on these results, we can find that KFNN is more sensitive to $\beta$ compared to $\alpha$. As $\beta$ tends to 1, KFNN tends to achieve optimal performance. Therefore, the default value of $\beta$ in this paper is set to 1. $\alpha$ hardly affects the performance of KFNN, which is set to 0.1 by default.

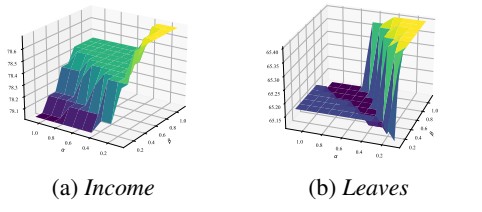

| (a) *Income* | (b) *Leaves* |

| (a) Macro-F1 score | (b) Class margin |

Figure 2: The Macro-F1 score (%) of KFNN on the *Income* and *Leaves* datasets when $\alpha$ and $\beta$ vary from 0.1 to 1.

Figure 3: The Macro-F1 score (%) and class margin (%) of KFNN or its components on the *Income* dataset.

**Ablation experiment.** There are two components in KFNN, namely label distribution enhancement (LDE) and K-free optimization (KF). To validate their effectiveness, we observe the Macro-F1 score of KFNN after taking away each component on the *Income* dataset. For simplicity, we use "KFNN-KF" to denote the variant of KFNN after taking away the component KF. Similarly, we create its another two variants "KFNN-LDE" and "KFNN-KF-LDE". Based on the results shown in Figure 3a, it can be seen that the performance becomes worse when any component is taken away. These results validate the effectiveness of LDE and KF. Figure 3b shows the change of the class margin before and after using our designed Kalman filter (observed on the first instance of *Income*). As can be seen from Figure 3b, compared to the margin before filter ($\widetilde{\mathcal{M}}_k$), the filtered margin ($\hat{\mathcal{M}}_k$) changes smoother. These results validate the effectiveness of our designed Kalman filter, which successfully mitigates the impact of noise incurred by neighbor instances.

## 6 Conclusion and future work

To ensure that each instance in crowdsourced datasets has a free neighborhood size, we propose a novel algorithm called KFNN. KFNN consists of two key components, namely label distribution enhancement and K-free optimization. Label distribution enhancement fuses the information from the attribute space and the multiple noisy label space. K-free optimization automatically determines the optimal neighborhood size for each instance by the max-margin learning. Both theoretical analysis and experimental results validate the effectiveness and robustness of KFNN.

Nevertheless, there are still some limitations in KFNN that can be improved in the future. For example, the parameters $\alpha$ and $\beta$ in the Kalman filter designed by KFNN can not automatically adapt to the dataset, which restricts the robustness of KFNN. In addition, in Eq. (4), transforming the distance distribution into the potential label distribution using max-min normalization is rough. Considering that the distance metric is not effective across all datasets (e.g., *autos* and *breast-cancer* in Table 1), this transformation may also lead to KFNN performing poorly. In the future, we will design more sophisticated parameters and transformations to improve KFNN.

## Acknowledgment

The work was partially supported by National Natural Science Foundation of China (62276241), Foundation of Key Laboratory of Artificial Intelligence, Ministry of Education, P.R. China (AI2022004), and Science and Technology Project of Hubei Province-Unveiling System (2021BEC007).

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

## Appendix A    The proof of Theorem 1

*Proof.* The covariance matrix $\mathcal{C}_q$ is a symmetric matrix. Therefore, if Assumption 1 holds, i.e., the covariance matrix $\mathcal{C}_q$ is a nonsingular matrix, $\mathcal{C}_q$ must be also a $M$-order symmetric matrix. This means that we can obtain $M$ different eigenvalues and $M$ mutually orthogonal normed eigenvectors when $\mathcal{C}_q$ is given. These orthogonal normed eigenvectors can form an orthogonal matrix $\mathcal{P}$, and thus $\mathcal{P}$ satisfies $\mathcal{P}^{-1}\mathcal{C}_q\mathcal{P} = \mathcal{P}^T\mathcal{C}_q\mathcal{P} = \Lambda$. Here, $\Lambda$ is a diagonal matrix with all $M$ eigenvalues of $\mathcal{C}_q$ as its elements of the principal diagonal. Moreover, the order of eigenvalues in $\Lambda$ should correspond to the order of eigenvectors in $\mathcal{P}$. □

## Appendix B    The proof of Theorem 3

*Proof.* Theorem 3 holds when the distance metric can work effectively given a crowdsourced dataset. The distance metric works effectively, which means that the smaller the $d(\boldsymbol{x}_1, \boldsymbol{x}_2)$, the more similar $\boldsymbol{x}_1$ and $\boldsymbol{x}_2$ are to each other and the more likely they are to belong to the same class. Therefore, in the neighbor sequence $< \boldsymbol{x}_i^1, \ldots, \boldsymbol{x}_i^k, \ldots, \boldsymbol{x}_i^N >$ of $\boldsymbol{x}_i$, $\boldsymbol{x}_i^k$ and $\boldsymbol{x}_i$ are more likely to belong to the same class when $k$ is small. At this point, when $\boldsymbol{P}_i^k$ is updated, the probability corresponding to the unknown true label $y_i$ of $\boldsymbol{x}_i$ will increase. When $k$ gradually increases and exceeds a certain threshold, $\boldsymbol{x}_i$ and $\boldsymbol{x}_i^k$ begin to belong to different classes, at which point the probability corresponding to $y_i$ will decrease. In other words, as $k$ increases from 0, the probability corresponding to $y_i$ increases first. As $k$ exceeds a certain threshold (the optimal neighborhood size), the probability corresponding to $y_i$ begins to decrease. Therefore, $max(\boldsymbol{P}_i^k)$ tends to be the probability corresponding to $y_i$ when $k$ increases from 0, and $k$ tends to be $K_i^*$ when $\hat{\mathcal{M}}_k$ achieves the highest value. □

## Appendix C    More descriptions of the experimental datasets and metrics

**Simulated datasets.**    The descriptions of the whole 34 simulated datasets are listed in Table 5. Here, "#Instances" denotes the number of instances, "#Attributes" denotes the number of attributes, "#Classes" denotes the number of classes, "Missing" denotes whether the dataset contains missing values and "Attribute type" denotes the type of attributes the dataset contains. These datasets are collected from different application scenarios and represent different crowdsourcing requirements.

**Real-world datasets.**    The *Income* dataset is annotated by 67 workers through the online platform Amazon Mechanical Turk (AMT), and each instance is annotated by 10 different workers. The *Income* dataset is a binary crowdsourced dataset, which contains 600 instances, 6000 labels, 10 attributes (nominal attributes) and 0 missing values. The *Leaves* dataset is annotated by 83 workers through AMT, and each instance is annotated by 10 different workers. The *Leaves* dataset is a multi-class crowdsourced dataset, which contains 384 instances, 3840 labels, 64 attributes (numeric attributes) and 0 missing values.

**Experimental metrics.**    The integration accuracy is calculated as follows:

$$Accuracy = \frac{\sum_{i=1}^N \delta(\hat{y}_i, y_i)}{N}. \tag{17}$$

The Macro-F1 score is calculated as follows:

$$F1 = \frac{\sum_{q=1}^Q \frac{2*Precision_q*Recall_q}{Precision_q+Recall_q}}{Q}, \tag{18}$$

where $Precision_q$ and $Recall_q$ can be calculated as follows:

$$Precision_q = \frac{\sum_{i=1}^N \delta(\hat{y}_i, c_q) * \delta(y_i, c_q)}{\delta(\hat{y}_i, c_q)}. \tag{19}$$

$$Recall_q = \frac{\sum_{i=1}^N \delta(\hat{y}_i, c_q) * \delta(y_i, c_q)}{\delta(y_i, c_q)}. \tag{20}$$

Table 5: The descriptions of 34 simulated datasets.

| Dataset | #Instances | #Attributes | #Classes | Missing | Attribute type |
|---------|-----------|-------------|----------|---------|----------------|
| anneal | 898 | 38 | 6 | yes | hybrid |
| audiology | 226 | 69 | 24 | yes | nominal |
| autos | 205 | 25 | 7 | yes | hybrid |
| balance-scale | 625 | 4 | 3 | no | numeric |
| biodeg | 1055 | 41 | 2 | no | numeric |
| breast-cancer | 286 | 9 | 2 | yes | nominal |
| breast-w | 699 | 9 | 2 | yes | numeric |
| car | 1728 | 6 | 4 | no | nominal |
| credit-a | 690 | 15 | 2 | yes | hybrid |
| credit-g | 1000 | 20 | 2 | no | hybrid |
| diabetes | 768 | 8 | 2 | no | numeric |
| heart-c | 303 | 13 | 5 | yes | hybrid |
| heart-h | 294 | 13 | 5 | yes | hybrid |
| heart-statlog | 270 | 13 | 2 | no | numeric |
| hepatitis | 155 | 19 | 2 | yes | hybrid |
| horse-colic | 368 | 22 | 2 | yes | hybrid |
| hypothyroid | 3772 | 29 | 4 | yes | hybrid |
| ionosphere | 351 | 34 | 2 | no | numeric |
| iris | 150 | 4 | 3 | no | numeric |
| kr-vs-kp | 3196 | 36 | 2 | no | nominal |
| labor | 57 | 16 | 2 | yes | hybrid |
| letter | 20000 | 16 | 26 | no | numeric |
| lymph | 148 | 18 | 4 | no | hybrid |
| mushroom | 8124 | 22 | 2 | yes | nominal |
| segment | 2310 | 19 | 7 | no | numeric |
| sick | 3772 | 29 | 2 | yes | hybrid |
| sonar | 208 | 60 | 2 | no | numeric |
| spambase | 4601 | 57 | 2 | no | numeric |
| tic-tac-toe | 958 | 9 | 2 | no | nominal |
| vehicle | 846 | 18 | 4 | no | numeric |
| vote | 435 | 16 | 2 | yes | nominal |
| vowel | 990 | 13 | 11 | no | hybrid |
| waveform | 5000 | 40 | 3 | no | numeric |
| zoo | 101 | 17 | 7 | no | hybrid |

