# OpenReview forum: "KFNN: K-Free Nearest Neighbor For Crowdsourcing"
_NeurIPS.cc/2024/Conference — NeurIPS 2024 poster_

### Official Review · Reviewer_4Kyj · 2024-07-08

**Soundness:** 3
**Presentation:** 3
**Contribution:** 3
**Rating:** 7
**Confidence:** 4

**Summary:**

This paper proposes a novel algorithm, KFNN (K-free Nearest Neighbor), which is specifically designed to enhance label integration for crowdsourcing. KFNN integrates two key components named label distribution enhancement and K-free optimization, which significantly contribute to improving the effectiveness and robustness of the label integration process. The idea of automatically determining the optimal neighborhood size for each instance is particularly innovative and well-executed. The experimental results further validate the effectiveness and robustness of the proposed algorithm.

**Strengths:**

1.	The KFNN proposed in this paper is interesting and innovative. The authors reveal the limitations of fixed neighborhood sizes in existing label integration algorithms and propose an algorithm that automatically determines the optimal neighborhood size based on instance attributes and noisy labels. This algorithm significantly improves the robustness of label integration.
2.	The paper provides a solid theoretical foundation for the proposed KFNN algorithm, followed by comprehensive experimental validation. The theoretical analysis is robust and convincingly demonstrates the expected performance improvements. The experiments are well-designed and cover a wide range of datasets, both simulated and real-world, to ensure the generalizability of the results. The experimental results, including comparisons with baseline algorithms, further validate the effectiveness and robustness of the proposed algorithm.
3.	The paper is well-written and clearly presents the proposed methodology and findings. The structure of the paper is logical, making it easy to follow the complex concepts introduced. The use of figures and tables to illustrate key points is effective and aids in comprehension.

**Weaknesses:**

1. While the paper provides strong theoretical and experimental results, there is limited discussion on the computational efficiency and scalability of the proposed KFNN algorithm. I suggest moving the algorithmic flow and time complexity analysis from Appendix A to the main text.
2. There are some repetitive sentences and structures in this paper that should be further condensed. For example, Sections 5.1 and 5.2 should be merged and the repetitive statements in them should be deleted.
3. The experiments are already comprehensive, but analysis and discussion of the optimal neighborhood size determined by KFNN could still be added, which would help to understand how the neighborhood size should be set. Moreover, according to the results presented in Tables 1-4, KFNN is generally highly effective. However, on few datasets, KFNN does not perform as well as MV. These anomalies are valuable for identifying deficiencies in KFNN and should be further investigated and discussed.

**Questions:**

Please refer to the Weaknesses.

**Limitations:**

The authors openly discuss the limitations of their work, particularly the empirical parameters in the Kalman filter and the roughness of the distribution transformation process. Please refer to the Weaknesses for other limitations I have found.

---

> ### Author Rebuttal · Authors · 2024-08-04
>
> **Reviewer 4Kyj：**
>
> **Q1:** While the paper provides strong theoretical and experimental results, there is limited discussion on the computational efficiency and scalability of the proposed KFNN algorithm. I suggest moving the algorithmic flow and time complexity analysis from Appendix A to the main text.
>
> **Author Response:** Thanks for your valuable comments. Although our KFNN needs to determine an optimal neighborhood size for each instance, it is not a wrapper algorithm. Label integration does not divide the crowdsourced dataset into training, validation, and test sets. As a result, KFNN determines $K_i$ immediately when inferring $\hat{y}_i$ without a validation phase. Therefore, the computational efficiency and scalability of KFNN are comparable to existing KNN-related label integration algorithms. In the final version of the paper, we will move the algorithmic flow and time complexity analysis from Appendix A to the main text. Thanks again for your valuable comments.
>
> **Q2:** There are some repetitive sentences and structures in this paper that should be further condensed. For example, Sections 5.1 and 5.2 should be merged and the repetitive statements in them should be deleted.
>
> **Author Response:** Thanks for your valuable comments. Indeed, there are some repetitive sentences and structures in our paper. In the final version of the paper, we will merge Sections 5.1 and 5.2 and delete these repetitive sentences. Thanks again.
>
> **Q3:** The experiments are already comprehensive, but analysis and discussion of the optimal neighborhood size determined by KFNN could still be added, which would help to understand how the neighborhood size should be set. Moreover, according to the results presented in Tables 1-4, KFNN is generally highly effective. However, on few datasets, KFNN does not perform as well as MV. These anomalies are valuable for identifying deficiencies in KFNN and should be further investigated and discussed.
>
> **Author Response:** Thanks for your valuable comments. In our KFNN, the optimal neighborhood size for each instance depends on both its attributes and multiple noisy labels. Based on its attributes, we can calculate the distance between this instance to the corresponding subset of each class. If the distance of this instance to one class is much smaller than the distance to other classes, it tends to be close to the center of this class. At this time, its optimal neighborhood size tends to be larger. In addition, if the multiple noisy labels of the instance are highly consistent, it means that this instance is easily distinguishable. At this time, its optimal neighborhood size tends to be smaller. Indeed, KFNN does not perform as well as MV on a small number of datasets, such as autos, breast-cancer, and diabetes. Considering that other KNN-related algorithms (LAWMV and MNLDP) also typically perform poorly on these datasets, we believe that the reason for the poor performance of KFNN is that these datasets are not well suited for distance measures. In the final version of the paper, we will include these discussions on optimal neighborhood size and anomalous experimental results. Thanks again for your valuable comments.

---

> > ### Comment · Reviewer_4Kyj · 2024-08-12
> > **Response**
> >
> > Thanks for your reply. Please incorporate some of the discussion into the final paper.

---

### Official Review · Reviewer_B2kv · 2024-07-10

**Soundness:** 4
**Presentation:** 3
**Contribution:** 4
**Rating:** 8
**Confidence:** 5

**Summary:**

The paper presents a novel label integration algorithm, KFNN (K-Free Nearest Neighbor), designed to enhance the performance of crowdsourcing platforms by intelligently determining the optimal neighborhood size for each instance based on its attributes and noisy labels. The authors propose a two-component solution involving label distribution enhancement and K-free optimization, which leverages the Mahalanobis distance and a Kalman filter to mitigate noise from neighbor instances. The paper's claims are well-aligned with the theoretical and experimental results, demonstrating the effectiveness and robustness of KFNN against existing state-of-the-art algorithms in various crowdsourcing scenarios.

**Strengths:**

1.	Novel contribution to an important problem
The innovative approach of highlighting the limitations caused by fixed neighborhood sizes in existing label integration algorithms, and using attributes and noisy labels to determine the neighborhood size for each instance automatically, is a significant contribution to crowdsourcing.

2.	Complete and rigorous theoretical proof
The theoretical underpinnings are sound, with clear assumptions and proofs provided for the proposed methods. The use of the Mahalanobis distance and the Kalman filter is well-justified.

3.	Good writing quality and clarity
This paper is well-written and enjoyable to read. The challenges are clearly stated and the contributions are easy to capture.

4.	Reproducibility
The paper's open data and code policy is highly appreciated, promoting research transparency. Enhancing reproducibility with clear versioning and setup instructions would be a valuable addition, showcasing a strong commitment to open scientific practices.

**Weaknesses:**

1.	Simulation experiment results
The symbol • indicates that the algorithm in the row significantly outperforms the algorithm in the corresponding column. How is "significantly outperforms" defined for Macro-F1 score and integration accuracy?

2.	Ablation experiment results
Since this study focuses on automatically adjusting neighborhood sizes, how does the performance of this method compare with baselines that use fixed neighborhood sizes?

**Questions:**

see weaknesses

**Limitations:**

see weaknesses

---

> ### Author Rebuttal · Authors · 2024-08-04
>
> **Reviewer B2kv：**
>
> **Q1:** Simulation experiment results The symbol • indicates that the algorithm in the row significantly outperforms the algorithm in the corresponding column. How is "significantly outperforms" defined for Macro-F1 score and integration accuracy?
>
> **Author Response:** Thanks for your valuable comments. The Wilcoxon signed-ranks test evaluates the performance differences between two algorithms across multiple datasets by ranking the absolute values of their differences. Specifically, we first calculate the differences between the performance scores (Macro-F1 score or integration accuracy) of the two algorithms on the 34 datasets. Then, these differences are ranked according to their absolute values, with average ranks assigned in case of ties. Subsequently, Let $R^+$ and $R^-$ be the sum of ranks for positive and negative differences, respectively. They can be calculated as follows:
> $$
>   R^+ = \sum_{d_i>0}rank(d_i) + \frac{1}{2}\sum_{d_i=0}rank(d_i),
> $$
> $$
>   R^- = \sum_{d_i<0}rank(d_i) + \frac{1}{2}\sum_{d_i=0}rank(d_i),
> $$
> where $d_i$ denotes the difference between the performance scores of the two algorithms on $i$-th out of 34 datasets, $rank(d_i)$ denotes the rank of $d_i$. Next, let $T$ be the smaller of $R^+$ and $R^-$. From the table of exact critical values of the Wilcoxon test, it can be found that the exact critical values of $T$ at significance levels $\alpha$ = 0.05 and $\alpha$ = 0.1 are 182 and 200 when the number of datasets is 34, respectively. This means that the two algorithms are significantly different with $\alpha$ = 0.05 when $T$ is less than or equal to 182. At this time, the second algorithm significantly outperforms the first if $T$ is equal to $R^-$. The first algorithm significantly outperforms the second if $T$ is equal to $R^+$. Thanks again for your valuable comments.
>
> **Q2:** Ablation experiment results Since this study focuses on automatically adjusting neighborhood sizes, how does the performance of this method compare with baselines that use fixed neighborhood sizes?
>
> **Author Response:** Thanks for your valuable comments. Among the existing label integration algorithms, both LAWMV and MNLDP are baselines using a fixed neighborhood size. Our simulated and real-world experimental results demonstrate that KFNN significantly outperforms both MNLDP and LAWMV. Additionally, in our ablation experiment, KFNN-KF is the version of KFNN employing a fixed neighborhood size. It can be found from Figure 3(a) that KFNN performs better than KFNN-KF. These findings highlight the superior performance of KFNN compared to baselines using a fixed neighborhood size. Thanks again for your valuable comments.

---

> > ### Comment · Reviewer_B2kv · 2024-08-13
> >
> > Thanks for the author's reply, my concerns have been well explained. I will keep my score.

---

### Official Review · Reviewer_qVhb · 2024-07-11

**Soundness:** 3
**Presentation:** 4
**Contribution:** 3
**Rating:** 7
**Confidence:** 4

**Summary:**

This paper proposes a novel label integration approach KFNN by adaptively determining the optimal neighborhood size. KFNN utilizes a Mahalanobis distance distribution to model the relationship between each instance and all classes. The authors also provide adequate theoretical analysis to illustrate the effectiveness of the proposed method. Experiments demonstrate that the proposed method can achieve the state-of-the-art performance on simulation and real-world dataset. The paper is well-written and easy to follow. This idea is very intuitive and effective for crowdsourcing task. The paper proves the effectiveness of introducing Mahalanobis distance distribution for crowdsourcing from the perspective of methodology, theory and experiments.

**Strengths:**

1. The paper is well-written and easy to follow. The logic of the whole paper is clear.
2. The paper’s idea is very intuitive and effective for crowdsourcing task. The authors introduce the Mahalanobis distance distribution to model the relationship between each instance and all classes. Experiments verify that the proposed method can achieve the best performance compared with SOTAs.
3. The authors provide adequate evidences to verify the effectiveness of the proposed method from the perspective of methodology, theory and experiments on simulation and real-world datasets.

**Weaknesses:**

1. In section 2, the authors introduce two categories of label integration algorithms. And the proposed KFNN belongs to the algorithms which leverage neighbor instance. I suggest adding some discussion about the pros and cons of these two categories of approaches.
2. In methodology part and theoretical analysis part, the authors discuss the superiority of Mahalanobis distance compared with Euclidean distance. Can the authors verify the difference between Mahalanobis distance and Euclidean distance on this task from an experimental perspective?
3. In Table 3 and Table 4, why some results are missing? Appropriate explanation facilitates reading of the paper.

**Questions:**

Please refer to weakness. My biggest concern is the experiments for the comparison between Mahalanobis distance and Euclidean distance.

**Limitations:**

yes

---

> ### Author Rebuttal · Authors · 2024-08-04
>
> **Reviewer qVhb：**
>
> **Q1:** In section 2, the authors introduce two categories of label integration algorithms. And the proposed KFNN belongs to the algorithms which leverage neighbor instance. I suggest adding some discussion about the pros and cons of these two categories of approaches.
>
> **Author Response:** Thanks for your valuable comments. The first category of algorithms does not leverage neighbor instances, considering only the information of the instance itself or the information of all instances globally in label integration. While simpler and more efficient, these algorithms are limited in effectiveness because each instance can only obtain few noisy labels. The second category of algorithms performs label integration by leveraging information from neighbor instances obtained by the KNN algorithm, which improves performance by using additional information from neighbor instances. However, these algorithms all assume a fixed neighborhood size for each instance, which is often unrealistic and thus limits their effectiveness. In the final version of the paper, we will add a paragraph to section 2 discussing the pros and cons of these two categories of algorithms. Thanks again for your valuable comments.
>
> **Q2:** In methodology part and theoretical analysis part, the authors discuss the superiority of Mahalanobis distance compared with Euclidean distance. Can the authors verify the difference between Mahalanobis distance and Euclidean distance on this task from an experimental perspective?
>
> **Author Response:** Thanks for your valuable comments. The main reason we chose the Mahalanobis distance in KFNN is that it can directly measure the distance from an instance to a dataset. Meanwhile, the Mahalanobis distance does not suffer from the correlation and magnitude of attributes. To respond to this comment, we defined the Euclidean distance of an instance to the centroid of a dataset as the distance of the instance to this dataset, replacing Equation 3 in the paper. Equation 6 was directly replaced by the Euclidean distance between the two instances. We denote this version of KFNN as KFNN-Euc. Subsequently, we compared KFNN and KFNN-Euc on the Income dataset. The Macro-F1 score and integration accuracy of KFNN-Euc on the Income dataset are 50.35% and 52.33%, respectively, significantly lower than those of KFNN (78.69% and 77.17%) shown in Figure 1. These experimental results validate the superiority of the Mahalanobis distance in KFNN. In the final version of the paper, we will include KFNN-Euc in the ablation experiment to demonstrate the superiority of the Mahalanobis distance in KFNN. Thanks again for your valuable comments.
>
> **Q3:** In Table 3 and Table 4, why some results are missing? Appropriate explanation facilitates reading of the paper.
>
> **Author Response:** Thanks for your valuable comments. Tables 3 and 4 show the results of the Wilcoxon signed-rank test. In Tables 3 and 4, the symbol • indicates that the algorithm in the row significantly outperforms the algorithm in the corresponding column, and the symbol ◦ indicates the exact opposite of that indicated by the symbol •. Missing items indicate no significant difference between the algorithm in the row and the algorithm in the column. The significance levels of the lower and upper diagonals in Tables 3 and 4 are 0.05 and 0.1, respectively. For example, the Wilcoxon test result between MV and LAWMV on the upper diagonal of Table 3 is missing, indicating no significant difference between MV and LAWMV in terms of the Macro-F1 score when the significance level is 0.1. In the final version of the paper, we will include a detailed explanation of the missing items. Thanks again for your valuable comments.

---

> > ### Comment · Reviewer_qVhb · 2024-08-12
> > **Response for Rebuttals.**
> >
> > Thank you for the helpful response that addressed my concern.

---

### Official Review · Reviewer_MQSy · 2024-07-12

**Soundness:** 1
**Presentation:** 2
**Contribution:** 2
**Rating:** 3
**Confidence:** 4

**Summary:**

This paper introduces a new algorithm for label integration called KFNN. Existing methods related to KNN produce more noisy labels; however, they fix the neighborhood size, regardless of the fact that instances close to the center of classes should have more neighbors than instances close to the boundary of classes. To tackle this problem, KFNN estimates a Mahalanobis distance distribution between each instance and all classes to enhance the multiple noisy label distribution and utilizes a Kalman filter to mitigate the impact of noise. Finally, KFNN can automatically determine the optimal neighborhood size through max-margin learning.

**Strengths:**

S1. The paper studies an important problem.
S2. A new solution is proposed to tackle the problem.
S3. Experiments are conducted on several datasets.

**Weaknesses:**

W1. The motivations need more enhancements.
W2. Some technical details require more explanations.
W3. The application scope of the proposed method in crowdsourcing is limited.
W4. The performance improvement of the proposed method is unsatisfactory.
W5. Experiments are conducted in a simulation environment, which can be much simpler than a real-world crowdsourcing platform.

**Questions:**

D1. The paper focuses on the KNN-related methods for label integration. However, the introduction didn’t justify the motivation of this concentration with convincing proofs. For instance, the motivation is basically explained with the sentence, “to alleviate this problem, recent works have begun to focus on leveraging neighbor instances [1, 11, 12] …”. However, there are also alternatives for label integration, so why considers KNN-related instead of the other types of solutions? Besides, there are much more studies (eg [R1]) that also target on this problem, which should be carefully discussed their pros and cons. Otherwise, the motivation looks weak.

D2. From the perspective of crowdsourcing, the studied problem is closely related to “truth inference”. However, in the references, there are only two papers on this topic: [25] (published in 2016) and [26] (published in 2023). More studies, which can be easily found in Google Scholar or DBLP, should be reviewed and compared (if possible).

D3. It is a little unclear how the principle of employing the same neighborhood size can impact performance. Please give more explanations.

D4. In addressing the question of fusing information from the attribute space and the multiple noisy label space, this paper tends to take an average between the multiple noisy label distribution and the potential label distribution. However, it might be worth exploring the possibility of introducing a tunable parameter to achieve a more optimal balance between these two distributions, rather than relying solely on an equal (50%) average.

D5. The application scope of the proposed method in crowdsourcing is limited. In my opinion, the proposed KFNN can be only used in simple and micro tasks in crowdsourcing, there are many other kinds of tasks in a real-world crowdsourcing platform, such as ranking [R2], which is not considered in the problem setting. Yet, the title, “KFNN: K-Free Nearest Neighbor For Crowdsourcing”, is a little over-claimed. At least, the paper should explicitly define the application scope. More types of crowdsourcing task can be found in existing surveys [R3, R4] on crowdsourcing.

D6. The performance improvement of the proposed method is unsatisfactory.
(1) Although the average Macro-F1 score of KFNN is better than the compared baselines, it can be notably worse than some of the baselines in certain datasets (eg MNLDP on the anneal dataset). This pattern weakens the motivation, since it’s unclear whether the limitation of existing solutions has been well addressed or not.
(2) In Table 2, the integration accuracy of KFNN is lower than that of MNLDP. Besides, it can be also notably worse than some of the baselines in terms of the integration accuracy (eg LAGNN and LAWMV on the breast-cancer dataset).
(3) Based on the current experimental results, the effectiveness of the proposed solution KFNN is questionable.

D7. Although several datasets are conducted in the experimental study, existing work on truth inference in crowdsourcing (eg [R2, R5]) usually deploys their solution in a real-world platform, such as AMT, to verify the performance. Therefore, the setup of the experimental study can be simplifier and less practical than the real-world scenario.

References:
[R1] Adaptive Integration of Partial Label Learning and Negative Learning for Enhanced Noisy Label Learning. AAAI 2024.
[R2] Xi Chen et al. Pairwise ranking aggregation in a crowdsourced setting. WSDM 2013.
[R3] Guoliang Li et al. Crowdsourced Data Management: A Survey. IEEE TKDE 2016.
[R4] Hector Garcia-Molina et al. Challenges in Data Crowdsourcing. IEEE TKDE 2016.
[R5] Yudian Zheng et al. Truth Inference in Crowdsourcing: Is the Problem Solved? VLDB 2017.

**Limitations:**

Please refer to the weaknesses and questions.

---

> ### Author Rebuttal · Authors · 2024-08-04
>
> **Reviewer MQSy：**
>
> Thanks a lot for your comments. Please find our detailed responses to your seven questions as follows.
>
> **Q1:** First, our research focuses on label integration in crowdsourcing, which differs from other research domains such as noisy label learning (NLL). Crowdsourcing typically employs workers to assign multiple noisy labels to each instance (one instance corresponds to multiple noisy labels), and the label integration aims to infer the unknown true label of this instance from these multiple noisy labels. In contrast, NLL aims to train robust classifiers from datasets with a single noisy label per instance (one instance corresponds to one noisy label). Therefore, existing works outside of label integration, such as [R1], are not within the scope of this paper. Second, due to cost constraints, each instance in crowdsourcing typically obtains only few noisy labels, which restricts the performance of label integration. State-of-the-art KNN-related label integration algorithms address this problem directly and efficiently, forming the basis of our research. We have described this motivation in the introduction (lines 32-42) and will refine it further in the final version of this paper.
>
> **Q2:** In crowdsourcing scenarios, the technical term "label integration" and "ground truth inference" are all common and widely used. Reference [1] describes that ''Ground truth inference is defined as a process of estimating the true label of each example from its multiple noisy label set. If we only focus on the label itself, it is also called label integration.". In the context of our paper, these terms are synonymous, all referring to the process of inferring the true label of each instance from multiple noisy labels. In our paper, we have chosen "label integration" to represent this process. Meanwhile, in our related work, we have comprehensively surveyed existing works, introducing a total of 18 representative works in label integration, with the latest published in 2024.
>
> [1] Learning from crowdsourced labeled data: a survey. Artificial Intelligence Review, 2016, 46(4):  543-576.
>
> **Q3:** The goal of finding nearest neighbors for an instance is to find potential instances of the same class around this instance. However, the number of instances of the same class around each instance is naturally different. Instances near the class center are surrounded by more instances of the same class, so they need larger neighborhood sizes to collect sufficient labels from similar instances. In contrast, instances near the class boundary need smaller neighborhood sizes to avoid including too many instances from other classes. These reasons are mentioned in lines 111-115 of our paper, and we will provide a more detailed explanation in the final version.
>
> **Q4:** Indeed, introducing a tunable parameter can help achieve a more optimal balance between the multiple noisy label distribution and the potential label distribution. Originally, we devised a weighted version of Eq. (5) as follows:
> $$
>   p_{iq} = \frac{\lambda * p(c_q|x_i,D_q) + (1-\lambda) * p(c_q|L_i)}{\sum_{q=1}^{Q}[\lambda * p(c_q|x_i,D_q) + (1-\lambda)* p(c_q|L_i)]},
> $$
> where $\lambda$ is a tunable parameter to balance these two distributions. However, we found experimentally that KFNN can still achieve good results with $\lambda$ set to 0.5 (equal average). To keep KFNN simple and with fewer parameters, we currently used an equal average to eliminate $\lambda$.
>
> **Q5:** Indeed, crowdsourcing is a broad research domain that includes various tasks such as label integration, noise correction, ranking, and so on. Our paper specifically focuses on label integration and presents a novel algorithm for label integration. The current paper title was proposed by referring to some classical works in label integration such as [2], hoping to inspire broader tasks in crowdsourcing beyond just label integration. In the final version of the paper, we will explicitly define our application scope by renaming our title to "KFNN: K-Free Nearest Neighbor for Label Integration in Crowdsourcing".
>
> [2] Community-based bayesian aggregation models for crowdsourcing. WWW `14, 2014: 155-164.
>
> **Q6:** Our proposed KFNN is not restricted to specific crowdsourcing scenarios, so we conducted experiments on the whole 34 datasets published by the CEKA platform. However, no label integration algorithm can achieve the best performance on all datasets. Therefore, we performed the Wilcoxon signed-rank test to further compare each pair of algorithms. The results in Table 3 show that KFNN significantly outperforms existing state-of-the-art label integration algorithms in terms of the Macro-F1 score. Additionally, as described in lines 254-258, we used the Macro-F1 score instead of integration accuracy as the main experimental metric. This is because the Macro-F1 score better reflects the performance of algorithms across different classes, while integration accuracy may not accurately reflect the performance on class-imbalanced datasets. Nevertheless, the results in Table 4 still show that KFNN achieves better or comparable integration accuracy compared with existing state-of-the-art algorithms. These results and analyses demonstrate the effectiveness and robustness of KFNN.
>
> **Q7:** Due to the double-blind policy, we did not deploy our solution in a real-world platform, such as AMT, and only submitted our code as a supplemental material. To verify the performance of KFNN in real-world crowdsourcing scenarios, as done in [R2, R5], we compared KFNN with existing state-of-the-art label integration algorithms using datasets Income and Leaves collected from the real-world crowdsourcing platform AMT. The results shown in Figure 1 are sufficient to demonstrate the effectiveness of the KFNN in real-world crowdsourcing scenarios. In the final version of the paper, we will provide a more detailed description of the collection of datasets Income and Leaves from the AMT platform.

---

> > ### Comment · Reviewer_MQSy · 2024-08-12
> > **Response to the author feedback**
> >
> > Dear authors,
> >
> > I have read the rebuttal, and thank you for considering my suggestions. Some of my concerns are well addressed, and the others aren't:
> >
> > (1) Q1 asks whether there are alternatives for label integration that can be used in crowdsourcing. If so, please discuss. Otherwise, please clarify that there are no such alternatives.
> >
> > (2) Q2: thank authors for acknowledging that these two terms are synonymous. That's why I have asked whether there are alternatives for the studied problem. As reviewed in the seminal survey work published in [R3], there are other options for this problem. Based on this reason, I have asked the authors to enhance the motivation of using KNN-based method instead of the others in Q1.
> >
> > (3) Q3 is generally satisfactory.
> >
> > (4) The claims for Q4 should be verified through experimental evaluations.
> >
> > (5) Q5 asks to clarify the type of crowdsourced task. Notice that, a ranking task in crowdsourcing still requires label integration (or ground truth inference). Thus, it is important to clearly define the application scope. From the current form of submission, the proposed method might be limited to only limited tasks types instead of general tasks in crowdsourcing. If this is the fact, the title and some other contents may need to be refined to avoid over-claiming.
> >
> > (6) Q6: as shown in the experimental results (Tables 1 and 2), the proposed method could perform worse than the selected baselines in either Macro-F1 score and integration accuracy under certain datasets by a large margin. Based on the current results, it seems that (1) different methods have their own pros and cons, and (2) the proposed method is not always the optimal among the compared ones.
> >
> > (7) Q7: thank you for acknowledging that the proposed method has not been verified by a real-world platform like AMT. Existing studies on crowdsourcing often conduct evaluations on a real-world platform, such as CrowdFlower (see Section 5.2 in [R2]), since real-world scenarios are more complex than a simulated experiment. Besides, the evaluation can be conducted under the double blind policy. Based on the current evaluations, it is unclear whether the proposed method can be effectively integrated in a real-world platform or not.
> >
> > Overall, I appreciate the efforts made in the rebuttal. I will take your responses into consideration when making my final decision.
> >
> > Best regards,

---

> > > ### Author Response · Authors · 2024-08-13
> > >
> > > **Reviewer MQSy：**
> > >
> > > **(1):** Q1 asks whether there are alternatives for label integration that can be used in crowdsourcing. If so, please discuss. Otherwise, please clarify that there are no such alternatives.
> > >
> > > **(2):** Q2: thank authors for acknowledging that these two terms are synonymous. That's why ... method instead of the others in Q1.
> > >
> > > **Author Response for (1) and (2):** Thanks for your valuable explanations. According to [R3] mentioned by the reviewer, there are alternatives for label integration (also known as answer aggregation or ground truth inference) in crowdsourcing to control the quality of crowdsourced datasets. These include worker modeling, worker elimination, and task assignment. Worker modeling characterizes the quality of workers, worker elimination eliminates low-quality workers and spammers, and task assignment assigns informative tasks to high-quality workers. However, these alternatives do not fully alleviate the impact of insufficient labels in label integration. Additionally, reminded by the comments of the reviewer, we will clarify our motivation and refine the title to "KFNN: K-Free Nearest Neighbor for Label Integration in Crowdsourcing" in the final version of the paper. Thanks again for your valuable comments.
> > >
> > > **(3):** Q3 is generally satisfactory.
> > >
> > > **Author Response for (3):** Thanks for your appreciative comments.
> > >
> > > **(4):** The claims for Q4 should be verified through experimental evaluations.
> > >
> > > **Author Response for (4):** Thanks for your valuable explanations. To verify our claims for Q4, we conducted experiments on the real-world crowdsourced dataset Income. According to the weighted version of Eq. (5) provided in the rebuttal, we set $\lambda$ to 0.1, 0.3, 0.5, 0.7, and 0.9, respectively, and then observed the performance of KFNN on the Income dataset. The experimental results are as follows:
> > > | |λ=0.1|λ=0.3|λ=0.5|λ=0.7|λ=0.9|
> > > |--|--|--|--|--|--|
> > > |F1|77.95|78.07|78.69|77.93|76.19|
> > > |Accuracy|76.33|76.50|77.17|75.83|73.33|
> > > | |
> > >
> > > From these results, it can be found that KFNN achieves optimal performance with $\lambda$ set to 0.5 (equal average). Therefore, these experimental results support our claims for Q4. Thanks again for your valuable comments.
> > >
> > > **(5):** Q5 asks to clarify the type of crowdsourced task. Notice that, a ranking task ... may need to be refined to avoid over-claiming.
> > >
> > > **Author Response for (5):** Thanks for your valuable explanations. According to [R3] mentioned by the reviewer, crowdsourced task types include single choice, multiple choice, rating, clustering, and labeling. Our current version of KFNN can be used for both single choice tasks and labeling tasks. We will clarify the application scope of KFNN in the final version of the paper. Furthermore, we plan to expand KFNN to other task types in the future. Thanks again for your valuable comments.
> > >
> > > **(6):** Q6: as shown in the experimental results (Tables 1 and 2), the proposed method ... the optimal among the compared ones.
> > >
> > > **Author Response for (6):** Thanks for your valuable explanations. Indeed, the current results show that under certain datasets (1) different methods have their own pros and cons, and (2) the proposed method is not always the optimal among the compared ones. These findings are normal and do not negate the effectiveness of our KFNN. KFNN is not restricted to specific crowdsourcing scenarios, so we conducted experiments on the whole 34 datasets published by the CEKA platform. In fact, no label integration algorithm can always achieve the best performance on all these datasets. Therefore, we performed the Wilcoxon signed-rank test to further compare each pair of algorithms. The statistical test strongly validates the effectiveness of KFNN. Thanks again for your valuable comments.
> > >
> > > **(7):**  Q7: thank you for acknowledging that the proposed method has not been verified ...  integrated in a real-world platform or not.
> > >
> > > **Author Response for (7):** Thanks for your valuable explanations. In the current version of the paper, both real-world and simulated experiments have been conducted to validate the effectiveness of KFNN. In our real-world experiments, we used the Income dataset and the Leaves dataset, which were collected from the real-world crowdsourcing platform AMT. The collection process of these two datasets is similar to that described in Section 5.2 in [R2]. The experimental results shown in lines 294-301 of our paper validate the effectiveness of KFNN in real-world crowdsourcing scenarios. Due to cost constraints, the number of real-world datasets is limited. Therefore, we also validated the effectiveness of KFNN through statistical tests on a large number of simulated datasets. The results in Tables 1-4 further validate the effectiveness of KFNN. In the final version of the paper, we will provide a more detailed description of the collection of datasets Income and Leaves from the AMT platform. Thanks again for your valuable comments.

---

> > > > ### Comment · Reviewer_MQSy · 2024-08-14
> > > > **Response to the rebuttal**
> > > >
> > > > Dear authors,
> > > >
> > > > I acknowledge your further responses. When discussing with the other reviewers on the final recommendation, I will consider the rebuttal.
> > > >
> > > > Thanks,

---

> > > > > ### Author Response · Authors · 2024-08-14
> > > > >
> > > > > Dear Reviewer MQSy,
> > > > >
> > > > > Thank you very much for your kindly reply and consideration.
> > > > >
> > > > > Thanks again.

---

> ### Author Response · Authors · 2024-08-12
>
> As the discussion period deadline nears, we would be deeply appreciative if you could kindly review our rebuttal and let us know if we have addressed your concerns. We’re more than happy to continue the conversation if you have any further questions. Thank you very much for your time and consideration.

---

### Decision · Program_Chairs · 2024-09-25

**Decision:**

Accept (poster)

**Comment:**

This paper proposes a novel label integration approach KFNN by adaptively determining the optimal neighborhood size. KFNN utilizes a Mahalanobis distance distribution to model the relationship between each instance and all classes.  The reviewers have reached a consensus to accept this paper.  Please follow the reviewers' suggestions to polish the main paper.